# Racial Democracy, Visibility, and the History of Colonial Brazilian Art

**Rachel A. Zimmerman**

Department of Art and Creative Media, Colorado State University Pueblo, Pueblo, CO 81001, USA; rachel.zimmerman@csupueblo.edu

**Abstract:** Since the nineteenth century, the history of colonial Brazilian art has highlighted the work of Afro-Brazilian men, specifically those with a white father and Black or parda mother. Antonio Francisco Lisboa, known as Aleijadinho, is the subject of countless books, exhibitions, television shows, and films. In addition to such famous men, dozens of other Afro-Brazilian artists are known and much scholarship has examined iconography and style with ties to African cultures. This extensive and important work has led to major exhibitions demonstrating Afro-Brazilian contributions as central to Brazil's past and present. Although intended as celebratory, the language and framing structures scholars use to discuss Afro-Brazilian artists from the colonial period are founded in white supremacy. The conception of Brazil as a nation where everyone is of mixed race, and therefore devoid of racism, is partly responsible for Aleijadinho's fame. This essay will clarify how this narrative of harmonious racial mixing and the focus on the visibly African perpetuates white supremacist interpretations of colonial Brazilian art and limits the study of Afro-Brazilian artists' work. I will propose ways to reframe workshop practice and improve connoisseurship using, among other cases, the lawsuit directed toward the white painter Manoel da Costa Ataíde that named the Afro-Brazilian artists who created one of his commissions. The essay builds on existing scholarship, acknowledging the violence of enslaving artists and promoting lines of inquiry that consider the agency and cultural positioning of Afro-Brazilian artists and patrons as subtle, ubiquitous, and heterogeneous.

**Keywords:** colonial Brazil; race; racial democracy; art history; hybridity; Manoel da Costa Ataide; Antonio Francisco Lisboa

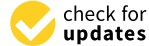



## 1. Introduction

In a speech given during the 52ª Assembleia Geral dos Povos Indígenas de Roraima on 13 March 2023, Brazil's president Luiz Inácio Lula da Silva commented on the transportation of enslaved Africans to Brazil: "All of the disgrace that this caused the nation, it caused a good thing, which is the mixture, the miscegenation."[1] Although this statement about slavery having a silver lining received negative attention, celebrating the mixing of races is a long-standing trope in Brazil that is frequently echoed among the general public, popular culture, and art historical scholarship. The most famous artist of Brazil's colonial period, a common household name, even among the public, is Antonio Francisco Lisboa, better known as Aleijadinho. His father was a white man and mother an enslaved African woman, and he has been the subject of countless books, articles, and exhibitions, as well as television shows and films. Many of the other well-researched artists of colonial Brazil were also of mixed race with a white father and Black or pardo mother (see Valerio 2023 in this special issue).[2]

Despite Lula's mention of slavery, discussions of Brazil's racially mixed population often gloss over the violence that initiated and fueled this racial mixing: white men sexually assaulting African and Indigenous women. In the 1930s, the sociologist Gilberto Freyre popularized the notion that Brazil was founded on sexual reproduction between white men and Black and Indigenous women. This, he argued, created a mixed-race society that was

free of racism, as opposed to the segregated United States. This conception of Brazil's racial dynamics has since become known as "racial democracy."[3] While Freyre imagined sex as a method for creating racial harmony, the use of sexual violence as a tool of colonization within Brazil has been studied for a long time (Aidoo 2018). In addition, the concept of racial democracy denies the persistence of white supremacist policies, institutions, structures, and culture in Brazil. In 1947, Black journalist and publisher Jose Correia Leite specified how the concept of harmonious racial mixing perpetuates anti-Blackness (Alberto 2011, pp. 202–3). Nonetheless, the idea of racial democracy has remained central to Brazil's mainstream culture, in part because it was employed during the military dictatorship from 1964 to 1985. The dictatorship used the narrative of racial democracy to deny the existence of racism and to repress racially marginalized groups (Alberto 2011, p. 245). Despite decades of activist efforts to dispel the ideology of racial democracy, a portion of art historical research about colonial Brazilian art continues to romanticize sexual relationships between white men and Black and parda women.

Brazil's colonial period spanned from 1500, when the first Portuguese ships arrived, to the early nineteenth century, when the royal family moved from Lisbon to Rio de Janeiro in 1808; Brazil was elevated from state to kingdom in 1815 and, finally, gained independence from Portugal in 1822. The field of art history, however, focuses predominantly on the eighteenth century to the 1820s, a period celebrated as the height of artistic achievement. This time period is often labeled the Brazilian Baroque, although it also includes Rococo and Neoclassical styles. Additionally, scholarship has predominantly focused on large-scale commissions, such as chapels, churches, and government building projects. Although Jose Luiz Mota Menezes' list of artists recorded in Pernambuco includes numerous people labeled "índio", art historical research on the Brazilian Baroque rarely discusses the contributions of Indigenous artists (Menezes 2002).[4]

In contrast, it is well-recognized that Afro-Brazilians, including master artists, free journeymen, and countless enslaved workers, were fundamental to Brazil's colonial artistic heritage. Founded on in-depth archival research and fanciful storytelling, the historiography of colonial Brazilian art history includes numerous monographs on pardo artists. Much research has also discussed visual traces of African religions within Catholic spaces and the depiction of Black and pardo people in colonial art. In addition, research on modern and contemporary Afro-Brazilian histories, religions, and arts is extensive. Emanoel Araujo's groundbreaking exhibition *A Mão Afro-Brasileira* (The Afro-Brazilian Hand) in 1988 and the exhibition *Histórias Afro-Atlânticas* (Afro-Atlantic Histories) in 2018 explored the complexities of Brazil's cultural heritage from the colonial period to the present.[5]

The scholarly emphasis on the contributions of Black and pardo artists is important and appropriate, given the quantity and quality of Afro-Brazilian involvement in the arts. Yet, because of the nearly ubiquitous narrative of the harmonious mixing of races, discussions of colonial art are frequently fraught with assumptions that are founded on white supremacist ideology and with an inconsistent approach to the violence of the system of slavery. In connection with a romanticized view of racial interactions, the art historical focus on visibility further leads to interpretations that are not entirely evidence-based. These approaches disguise the violence of the past, as well as the continuation of racism in the present. This article will examine how the romanticization of sexual assault distorts the interpretation of colonial artworks, how equating an artist's race with the visual appearance of their work is misleading, and how the legal and social circumstances of artistic production affected Afro-Brazilian artists. I will conclude with avenues of research that avoid harmful assumptions about race and instead employ more comprehensive interpretive models that acknowledge the heterogeneity of the Afro-Brazilian population.

As anthropologists Aisha M. Beliso-De Jesus and Jemima Pierre explain, "one does not have to *be* explicitly racist to reproduce white supremacy or its discursive formations" (Beliso-De Jesus and Pierre 2020, p. 71). The distinction between personal intent and white supremacist structures is crucial to this article, as art historical writings on Blackness in colonial Brazilian art are often intended to be anti-racist. Affirming that many of the

nation's most celebrated artworks were created by Afro-Brazilians and embracing the nation's African cultural heritage acts as a rejection of efforts to whiten Brazil's population and culture.[6] Like Freyre's attempt to distinguish Brazil from the segregated United States, President Lula's statement served to distance himself from his predecessor Jair Bolsonaro's overtly white supremacist platform.[7]

White supremacy is the belief that white people are superior to other races and are, therefore, justified in displacing, dominating, and enslaving non-white peoples, as well as the systems, institutions, and laws that maintain the ideological and material privileges of whiteness. Because nations throughout the Americas are founded on white supremacist structures and cultures, recognizing the white supremacist roots of art historical methodologies can be challenging. Many scholars, such as Mario de Andrade, Marianno Carneiro da Cunha, Tania Tribe, Trindade, Jaelson Bitran, and the countless researchers who have written about Antonio Francisco Lisboa, discuss how discriminatory laws and workshop practices restricted the activities of Afro-Brazilian artists, as well as the racist insults they faced from other artists and patrons. In recent decades, scholars have continued to revise and refine the understanding of Brazil's colonial artists with further archival research and additional interpretive methods.

Despite this recognition of systemic and cultural racism's barriers to Afro-Brazilian artists, the ideology of racial democracy and the white supremacist foundations of the field of art history continue to inform how some scholars frame their discussions of colonial art. Colonial artists and patrons perceived the paintings, sculptures, and architecture in their surroundings within the context of a dominant culture of normative whiteness, rather than a culture that celebrated the mixing of races. Archival documents, typically produced within legal and religious contexts, reveal normative whiteness in their inclusion of race for non-white, but not white, people. Baptismal records, for instance, typically mention whether the parents and godparents were enslaved, freed, pardo, or preto, but rarely labeled white individuals as such.[8] Although Brazil had a significant population of mixed-race individuals by the end of the colonial period, this was often a source of anxiety for white officials and community members, rather than a source of celebration. For instance, white men were typically discouraged from marrying women of color (Dantas 2009, pp. 125–29). Whiteness was not solely a function of ancestry, but was tied to socioeconomic and legal status. Thus, a person who had been recorded as pardo earlier in life could become white as their social standing increased (Dantas 2009, p. 126). This is significant for the discussion of artistic depictions of race, as race was often indicated with clothing, jewelry, or other visual cues, while depictions of skin tone are not reliable markers of legal status. Thus, interpreting messages about race in colonial Brazilian artworks requires nuance.

## 2. Ataíde and Harmonious Racial Mixing

In the 2007 book on the white artist Manoel da Costa Ataíde, art historian Adalgisa Arantes Campos details Ataíde's relationships with free and enslaved Afro-Brazilian people in order to portray him as a benevolent, generous, and devout person. He had six children with a free parda woman named Maria do Carmo Raimunda da Silva, two of whom died in infancy (Campos 2007, pp. 74–76). Maria was about 21 and the artist about 47 years old when their first child was born (Campos 2007, p. 77). Campos argues that the two had a loving relationship, an idea that would require substantially more evidence than is available to support (Campos 2007, p. 78). Claiming that the partnership was, from the onset, consensual and based on love, rather than coercion, overlooks the racial dynamics at play. As Erin L. Thompson succinctly states, "a yes doesn't mean yes when you can't say no" (Thompson 2018). Ataíde was wealthy, from a well-connected family, and held the prestigious military title of *alferes* (ensign). In a race-based slave society, a young Black woman would not have been able to safely decline the sexual advances of such a powerful white man. While scholars today cannot know how Ataíde's sexual relationship was initiated, the available evidence about race, age, and social standing suggests that Ataíde held considerable power over the young Maria do Carmo.

For decades, scholars have claimed that the loving relationship between Ataíde and the young parda Maria do Carmo was made visible in Ataíde's ceiling painting of the Assumption of the Virgin in the church of São Francisco de Assisi in Ouro Preto (Figure 1). Ataíde's rendition of the Virgin Mary has been described as a parda woman modeled after Maria do Carmo and the many angels as inspired by the artist's pardo children. The interpretation of the painting as a visual reflection of the artists' love for a mixed-race woman arose when the central medallion was covered in layers of grime that caused the painting to appear dark and largely monochromatic. As Beatriz Coelho, one of the restorers who worked on conservation of the ceiling from 1985 to 1988, explains "A thin layer of dark brown powder covered all of the medallion, including the figures of Our Lady of Porciuncula, King David and the angels," in addition to a layer of darkened varnish.[9] This discoloration had led scholars to believe that the medallion, unlike the rest of the ceiling, was painted in oil and was dominated by a strong chiaroscuro and muted sepia tones (Del Negro 1958, pp. 52–53; Frota 1982, p. 52). The color reproductions of the ceiling in Del Negro's 1958 book and Frota's 1982 book suggest that the brown discoloration was quite severe, with the angel's bodies appearing almost the same colors as the clouds and the Virgin's blue mantle almost black. Today, the ceiling is more cohesive with vibrantly colored tempera on both the medallion and surrounding faux architecture. The Virgin's skin tone is light and her garments feature bright red and blue. There is still a strong contrast between the brown and gray shadows and the highlights, but the chiaroscuro is overall less dark.

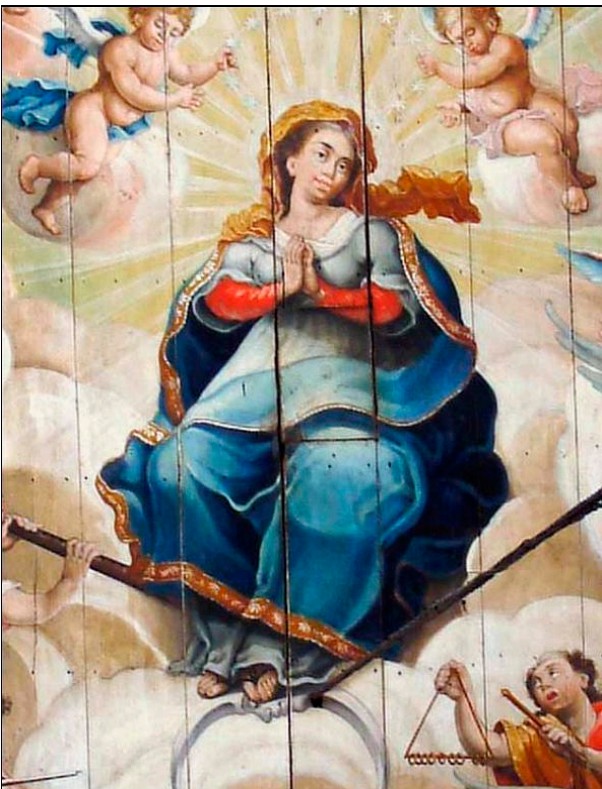

**Figure 1.** Detail, Manoel Costa da Ataide, Virgin Mary, ceiling of the church of São Francisco de Assis, Ouro Preto. Courtesy of Tetraktys.

Creating a painted figure that is recognizable as mixed race depends on the social construction of race and the artistic conventions for representing race as understood by both the artist and the intended viewers. Scholars describing Ataíde's Virgin as parda rarely explain what visual conventions within the painting communicate the Virgin's race. For instance, art historian Edward J. Sullivan states, the Virgin's "features clearly indicate

someone of African descent," but does not describe those features (Sullivan 2006, p. 51). In a discussion of another artist's paintings, Carla Mary S. Oliveira states, "the facial features of Mary approximate the mixed physical type, with full lips and undulating hair."[10] In contrast, Ataíde's Virgin, whom Oliveira also describes as representing a parda woman, has a small mouth and her hair covered. When comparing this painted Virgin with figures in Ataíde's other paintings, it is not clear that her facial features are intended to signify African descent. Most of Ataíde's painted faces include the same nose: straight with a slightly rounded tip; the same mouth: narrow and accentuated, with an undulating curve on the upper lip; and the same chin with a horizontal crease. I would argue that this is Ataíde's standard method of painting faces and does not signify a specific race.

Identifying the intended race based on the color of the painted Virgin's skin is unreliable. At this time, Brazilian artists had not established conventions for portraying people of mixed race with a specific color of skin. For instance, the Luso-Italian military engineer Carlos Julião's depictions predominantly distinguish certain racial types by clothing, rather than skin color (Tenreiro 2007, pp. 18, 23; Silva 2010, pp. 15, 133). While his Black figures are dark, the white and mulatta figures mostly feature the same pale skin. Only one figure labeled a mulata has a skin tone that is different from the nearby white figures: the "mulata receiving a letter from her mistress" in the *Four Ports Panorama* is light-skinned, but slightly darker than the young boy who hands her the letter and the dancing woman next to her.

In addition to ambiguous visual evidence, the context in which Ataíde's painting was created and viewed suggests that he and his patrons did not perceive the painted Virgin Mary as mixed race. The church that contains the painting belongs to the lay confraternity of Saint Francis of Assis, a prestigious organization restricted to white members. Would such an organization have accepted the depiction of a parda Virgin Mary prominently displayed on the nave ceiling of their church? Some scholars circumvent the issue of context by suggesting that Ataíde's representation of a parda Virgin was "perhaps unintentional," the result of him unconsciously incorporating the features of Maria do Carmo into the painting (Tribe 1996, p. 75; Oliveira 2011, p. 100). However, as race is a social construct and pigment has no race, a painted representation only has a race if the artist and viewers attribute a race to it. More likely, Ataíde did not intend to paint a mixed-race Virgin and his patrons did not perceive the representation as mixed race. In a culture dominated by normative whiteness, a white Virgin Mary was considered standard and any deviation from this standard would have needed to be explicit in order to be interpreted as such.

Finally, the interpretation of this painting as a reflection of Ataíde's loving relationship with a parda woman depends on assumptions about the timeline of Ataíde's work in the church. While Ataíde received payments for various painting and gilding projects at the church of São Francisco from 1800 to 1812, the surviving receipts do not mention the nave ceiling (Menezes 2007a, pp. 24–25; Trindade 1951, pp. 405–10). While the date range for Ataíde's work at the church overlaps with the birth of one or two of his children, scholars do not know when Ataíde painted the Virgin and surrounding angels.[11] Ataíde first impregnated Maria do Carmo in the beginning of 1809, when she was 20 years old.[12] The central medallion of the ceiling may or may not have been completed by this time.

Given the evidence about the painting, the patrons, the artist's relationships, and the visual conventions of representing biblical figures, the most probable interpretation of this ceiling painting is that it was intended to portray a white Virgin Mary. While it is possible that Ataíde and at least some of his patrons perceived his painting as representing a parda woman, the narrative that has supported this interpretation is largely founded on ideals that are anachronistic to the painting's creation. Rather than having been created as a reflection of a society where the mixing of races was viewed as a cause for celebration, twentieth- and twenty-first-century viewers have transformed the painted Virgin into a parda in order to serve as an emblem of Brazil's racial democracy.

### 3. The Fallacies of Visibility

Ataíde's reputedly mixed-race Virgin is only one example of many in which scholars seek out and label aspects of colonial Brazilian artworks that signal Africanness or Blackness. Carolyn Dean and Dana Leibsohn explain how such an approach can distort evidence and the experiences of early modern peoples within the context of colonial Spanish America. They argue that the scholarly focus on the visibility of Indigeneity and the labeling of Indigenous visual elements in colonial artworks typically obscures the processes of colonization, coercion, enslavement, patronage, choice, and negotiation in which such artworks are created (Dean and Leibsohn 2003). This section explores how the art historical focus on visibility limits research and causes misconceptions. I will discuss issues that arise when creating a definition of African visual language; the heterogeneity of Afro-Brazilian artists, patrons, and viewers; and how associating Afro-Brazilian artists with depictions of Black and pardo people and with iconography related to Africa is misleading.

Identifying African elements of an artwork requires scholars to create a definition of African visual language that is distinct from European, or Euro-American, visual language. In discussing a similar phenomenon in scholarship that seeks to distinguish European and Indigenous elements in colonial Spanish American art, Dean and Leibsohn write: "hybridity is not so much the natural by-product of an 'us' meeting a 'them', but rather the recognition—or creation—of an 'us' and a 'them'" (Dean and Leibsohn 2003, p. 6). Given that Africa is the second-largest continent on Earth and home to over one thousand cultures, religions, and languages, categorizing certain elements of artworks created in colonial Brazil as African likely involves some degree of homogenization. As activist Carlos Negreiros remarked in 1975 about the common conception of Afro-Brazilian heritage: "it's as if they grabbed blacks in Africa, mixed them in a blender, and tossed them into Brazil under one label: slaves" (Alberto 2011, p. 272).

This categorization of visual elements as either African and Black or European and white is especially inappropriate in the context of the type of art that is studied most for the colonial period: Catholic religious art and architecture. Many of the Africans arriving in Brazil were already Catholic, not from forced conversion while enslaved, but because of the long history of Catholicism in Central Africa. As several of Cécile Fromont's publications examine, Catholicism was well-established in the Kingdom of Kongo, beginning with the conversion of King Afonso in 1491. In Fromont's recent edited volume, *Afro-Catholic Festivals in the Americas*, scholars have reinterpreted Afro-Brazilian festivals and visual imagery to consider the intertwined relationships between Catholicism and Central African heritage.[13] The watercolors of an Afro-Brazilian festival in Rio de Janeiro that Carlos Julião painted toward the end of the eighteenth century are prominent examples of images that have been dissected into European and African parts and presented as evidence of the mixed culture of Brazil (Figure 2). Cécile Fromont and Junia Ferreira Furtado reexamined these representations in light of Central African Catholicism (Fromont 2013, 2019; Furtado 2019). The paintings depict the procession of the Afro-Brazilian confraternity of Our Lady of the Rosary, a particularly notable instance in which distinguishing between European and African elements distorts how Afro-Brazilians themselves performed and understood the festival. This procession incorporates aspects of the Kongo *sangamento*, a performance that reenacts the incorporation of Catholicism into Kongo culture with European symbols of status, such as coats of arms and swords (Fromont 2013, pp. 188–89). Thus, contemporary scholars examining Julião's watercolors typically perceive items such as gowns and crowns as European, whereas the performers, some of whom were forcibly brought to Brazil from the Kingdom of Kongo, viewed such items as a connection to their homeland.

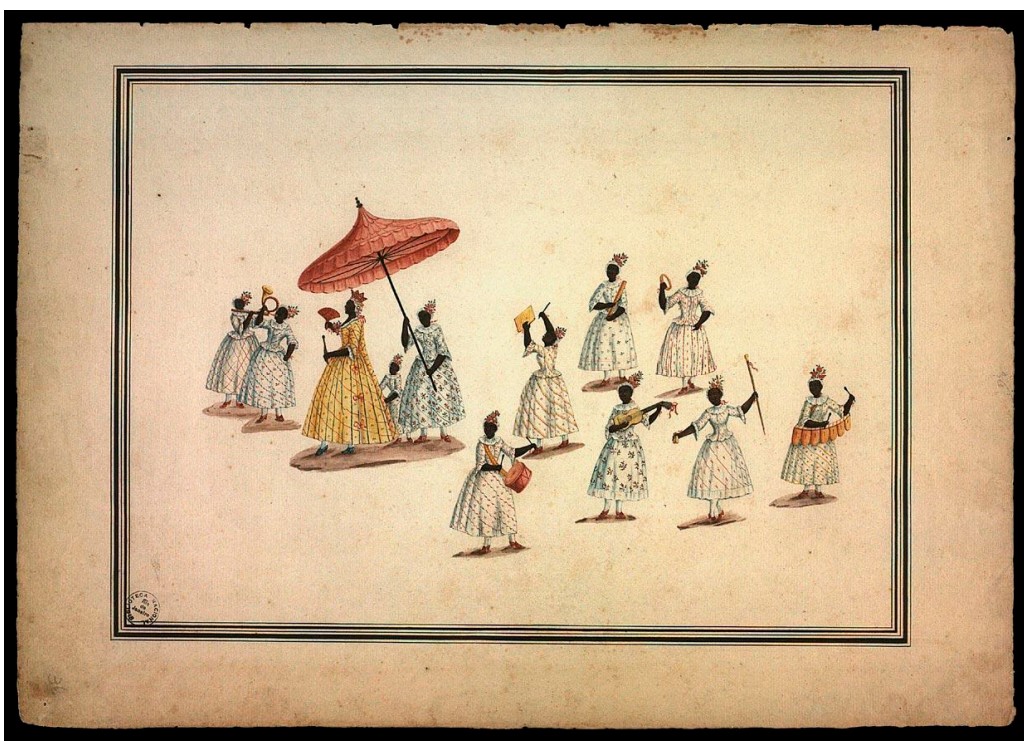

**Figure 2.** Carlos Julião, "Procession of the Queen," in Noticia Summaria do Gentilismo da Asia, late eighteenth century. Courtesy of Biblioteca Nacional, Brazil.

In contrast, people from West Africa, where Catholicism was not widespread, would have had a different experience of and relationship with Christian festivals and imagery. The survival and revival of aspects of Yoruba religious practices among Yoruba-descended Brazilians are well documented. In terms of art history, most research surrounding Yoruba or other African religious traditions being continued, adapted, and transformed in Afro-Brazilian contexts centers largely on small-scale objects confiscated by the police in the late-nineteenth and early twentieth centuries (Conduru 2022, p. 4; Buono 2015, pp. 30–31). Such objects, which the dominating white authorities perceived as subversive, stand in stark contrast to the churches that are the focus of discussions of the Brazilian Baroque. Commissioning and building churches involved the exchange of significant sums of money, government approval, and multiple levels of supervision from white officials. Although Afro-Brazilians maintained connections to their ancestral cultures, evidence for Yoruba-derived religious practices are less prevalent during the colonial period within the kinds of sources that art historians tend to use: written records and visual evidence from artworks.

Some scholars have, however, attempted to see Yoruba religious symbols in the decoration of churches belonging to Black and pardo confraternities, in particular in the church of Santa Efigênia in Ouro Preto. Lázaro Francisco da Silva identified many iconographic elements in the interior of the church of Santa Efigênia that he claimed were intended as references to Yoruba beliefs and evidence of subversive intentions among the confraternity members. For instance, he identified the letter "I" engraved in the chancel opposite the monograph IHS as a reference to "Ifá." He also described the carved cowry shells that project from the walls as doubling as vulvas that, from another angle, resemble phalluses. The combination of the phallus, vulva, and cowry shell relates to Yoruba mythology and ritual practice (Silva 1995, p. 75).

While Silva's articles have been frequently cited and his identification of iconographic elements within the church as related to Yoruba religion echoed, evidence that the builders, sculptors, and painters of the church and their patrons intended to make such clear connections is tenuous. As Jaelson Bitran Trindade points out, Silva's argument contains several flaws, such as the claim that the church's interior dates to the 1730s, when most confrater-

nity members had recently arrived from Africa. Archival records and stylistic evidence, instead, date the interior to the second half of the eighteenth century, when most members were likely to have been born in Brazil (Trindade 2019, pp. 62–64). Additionally, by this time, the majority of enslaved Africans brought to Minas Gerais were being captured in Central Africa, rather than West Africa (Kiddy 2005, pp. 41, 50). Silva himself acknowledges that several of the confraternity's officials in charge of the erection and adornment of the church were white (Silva 1995, p. 72). Some of the confraternity members were likely to have made connections between the church's iconography and the Yoruba religion, especially associating saints with Orishas. However, considering that the patrons were composed of Black and white confraternity officials, people born in Brazil, Central Africans, and West Africans, among others, and that several of the artists involved were white, it seems unlikely that the iconographic program of the church was deliberately designed to convey Yoruba messages that were subverting the Christian purpose of the church.

The scholarly focus on iconography that is related to Africa limits Afro-Brazilian visual culture. Even for those artists and patrons who were born in Africa and forcibly relocated to Brazil as adults, the artistic legacies that they left behind are the result of their lives in Brazil, negotiating between a range of social, political, cultural, and visual realities. Their experience of life in Brazil is no less real and no less part of themselves than their experience of life in Africa (Dean and Leibsohn 2003, p. 14). In some cases, aspects of Afro-Brazilian churches that do not visibly correlate to scholars' conceptions of Africa have not been granted thorough consideration.

When scholars identify iconography that appears related to African heritage, they often attribute the work to Black or pardo artists.[14] Scholars also frequently attempt to identify traces of an artist's race in their work when that artist is not white. The assumption that a Black or pardo artist's work will, by necessity, look African or contain Black figures denies artistic agency and the realities of artistic training and patronage, and is founded on white supremacist tropes. The association between an artist's race and their style and iconography stems from a modernist definition of art that Linda Nochlin famously critiqued in 1971 as "The naive idea that art is the direct, personal expression of individual emotional experience–a translation of personal life into visual terms" (Nochlin 1973, p. 5). Being an artist working on religious commissions in colonial Brazil involved artistic training, collaborators and apprentices, patrons, contracts, artistic conventions, religious dogma, and numerous other logistical considerations. Most of the colonial art and architecture subject to art historical study was commissioned by institutions, such as monastic communities, lay confraternities, and government offices. With such commissions, artists signed contracts that detailed the specifics of materials, styles, and iconography. Deviating from a contract or producing unsatisfactory work could lead to loss of pay, such as that which occurred in the case of Ataíde's commission for the chancel of the church of Nossa Senhora dos Pretos in Mariana, discussed below. Within this context, an artist's goal was not to express their racial identity, but to satisfy the patron, secure payment, and invite future patronage.

The idea of African influence pervading the work of Afro-Brazilian artists furthermore portrays artists as vessels of unconscious communication, denying Black and pardo artists agency. Examining how scholars describe the work of white artists in contrast with Black or Indigenous artists demonstrates that white artists are seen as exercising their talents of invention, making decisions about what and how they paint or sculpt. As participants in European colonization, white artists investigate and purposely adopt imagery and materials from other cultures. The work of Black and Indigenous artists, on the other hand, is viewed as bound to their race or culture of origin, however far removed in time and place, with traces of their race always expected to be visible in their art (Dean and Leibsohn 2003, p. 23).

Cases in which art historians reinterpret artworks after learning that they had been mistaken about the artist's race demonstrate the tenuous relationship between an artist's race and the visual appearance of an artwork. For instance, Adalgisa Arantes Campos saw evidence of Jose Gervasio de Souza's mixed race in his work when she believed that

he was a pardo artist born in Brazil. She described some of the figures in the altar of Nossa Senhora do Rosario dos Pretos in Ouro Preto as having pardo features (Campos 2002, p. 249). Once Campos had found documentary evidence that Gervasio de Souza was born in Portugal to presumably white parents, she departed from this approach of identifying signs of Blackness in his work and instead examined the religious iconography in his paintings (Campos 2012). She acknowledged that her desire to see Brazil's racial mixing visualized had clouded her interpretation of his paintings (Campos 2012, p. 3).

The depiction of Saint Gregory the Great as a Black man in the ceiling above the high altar in the church of Santa Efigênia in Ouro Preto has also been subject to varying interpretations and assumptions about the artist's race (Figure 3). Saint Gregory is shown with the other three doctors of the church (Figures 4 and 5). This iconography appears in several other churches in the region, but elsewhere, all four doctors are portrayed as white. When Jair Afonso Inácio cleaned and restored the painting in 1968, he not only remarked on the "Black pope", but also claimed that the three other figures were Black (Trindade 2019, p. 77). Lázaro Francisco da Silva described the three other doctors as having Black physical features, and Nancy Nery da Conceição, relying heavily on Silva, described them as pardos (Silva 1995, p. 70; Conceição 2016, p. 63). Other scholars have been more cautious, often only naming one of the other doctors as mixed race, such as Tania Tribe, who describes Saint Ambrose as mixed race (Tribe 1996, p. 75). How the distinction between white and mixed-race figures is being made is, however, unclear. All three figures have the same color skin on their faces and hands, full beards, similar facial features, and those without head coverings have the same wavy hair. In fact, some recent publications mention a Black Saint Gregory, but do not mention the race of the other doctors or describe them as white. Assumptions have also been made about the ceiling's painter. Silva suggests that not only the Black Saint Gregory, but also the sculpted iconographic elements that he associates with Yoruba religion, were potentially created by a Black artist, possibly "some slave who worked with his master," while Benatti, Teruya, and Maio propose that the Black Saint Gregory was painted by a different artist than the other doctors (Silva 1995, p. 71; Benatti et al. 2022, p. 15). Moreover, those scholars associating the Black Saint Gregory with rebellion, an enslaved artist, and with Chico Rei, the legendary king who is said to have funded the church of Santa Efigênia, often mistake the papal camauro for a Phrygian cap, a symbol of liberty (Silva 1995, p. 70; Conceição 2016, p. 57; Trindade 2019, p. 58).

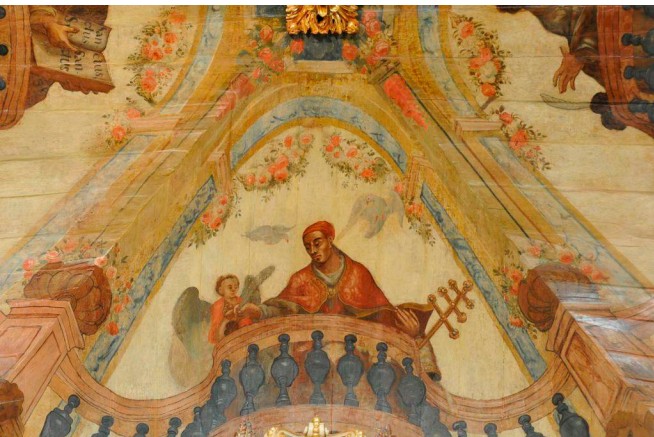

**Figure 3.** Saint Gregory the Great, detail of ceiling painting of the chancel of the church of Santa Efigênia, Ouro Preto. Courtesy of Rodrigo Coelho, eravirtual.org.

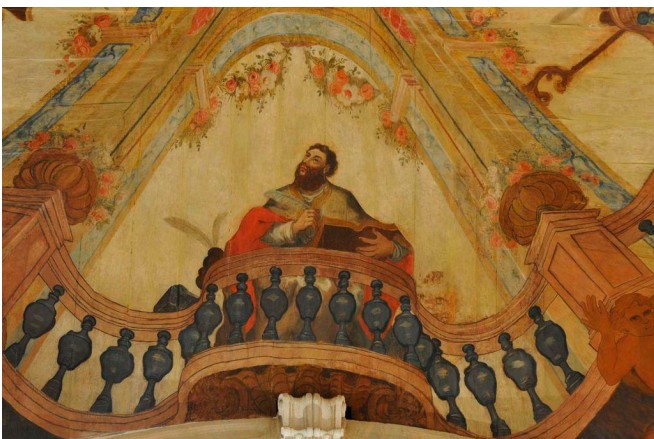

**Figure 4.** Saint Ambrose, detail of ceiling painting of the chancel of the church of Santa Efigênia, Ouro Preto. Courtesy of Rodrigo Coelho, eravirtual.org.

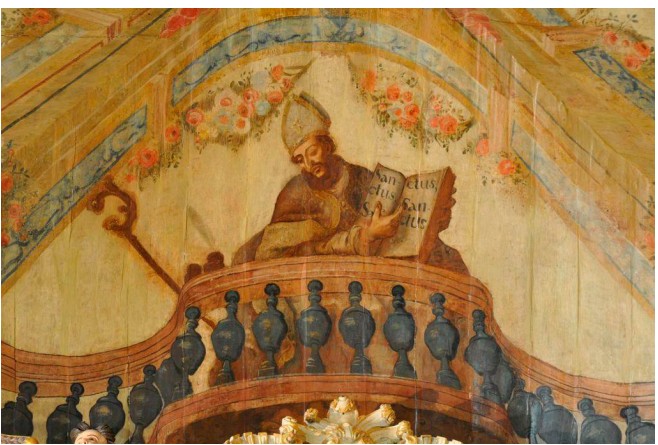

**Figure 5.** Saint Augustine, detail of ceiling painting of the chancel of the church of Santa Efigênia, Ouro Preto. Courtesy of Rodrigo Coelho, eravirtual.org.

As Dean and Leibsohn suggest in the case of the Spanish Americas, the art historical emphasis on the visibility of Indigenous iconography masks the violence of colonization (Dean and Leibsohn 2003, p. 14). In the context of Brazil, being able to see Yoruba symbols and pardo figures creates a more comfortable tale of the combination of cultures, the precursors to racial democracy. This focus on the visible not only leads scholars to perceive things that the artists and patrons probably did not see, but also leads to omissions. Some work by Black and pardo artists is understudied because it does not look Afro-Brazilian, according to the definition of Afro-Brazilian art that scholars have created.

## 4. Artistic Practice and Legal Structures

Close examination of archival records and artworks offers insights into the complex negotiations that took place throughout the erection and adornment of chapels, churches, and other artistic projects. Such projects often involved master artists who signed contracts and oversaw workshops, journeymen, apprentices, and enslaved artists. The racial dynamics within the group of artists working on a project could vary. This section explores the division of labor, the means of supervision that controlled artists' work, the available information about enslaved artists, and how the narrative of racial democracy leads to biased discussions of the relationships between masters and the artists they enslaved. For this discussion, the trial records from the 1826 lawsuit in which the white master painter Ataíde sued an Afro-Brazilian confraternity serve as a case study.

Fortunately, much archival documentation has survived to guide the study of colonial Brazilian art. Such records offer some answers about the roles that Afro-Brazilian artists were able to play within the context of projects that involved receipts and contracts. Major artistic commissions, such as the painting and gilding of church interiors, were carried out by groups of artists. White and pardo master artists whose names are recorded on contracts and receipts employed apprentices and assistants, both free and enslaved, and also subcontracted tasks to other artists (Trindade 2002; Santiago 2008). For the chancel ceiling of the church of Santa Efigênia, with its Black Saint Gregory, the commission and payment went to the white master artist Manuel Rabelo de Souza, although scholars have attributed the work to the white master painter Joao Batista de Figueiredo, himself a prestigious artist who held the military rank of captain and took on his own commissions (Trindade 2019, pp. 59–60).

Because such commissions involved considerable sums of money and multiple teams of workers, contracts were typically quite detailed in their descriptions of the work to be produced. Master artists were responsible for assuring the quality of the work of any apprentices or journeymen, and the completed work was subject to examination by third parties. How stringent contracts were depended on the relative importance of the parts of the commission. Often, the master artist focused on elements of a commission that were deemed most sacred, most visible, and most important while leaving the workshop to complete work that was deemed peripheral, such as backgrounds and areas of the nave, as opposed to the high altar (Santiago 2008, p. 81). While artists were permitted, and encouraged, to exercise creativity and invention, deviations from contracts had consequences. One such example survives in the records of a lawsuit. In 1823, Ataíde was commissioned to complete numerous tasks in the chancel and sacristy of the church of Nossa Senhora do Rosário dos Pretos in Mariana, including the ceiling painting, gilding, white washing, and painting the lower section of the chancel walls in imitation of azulejos.[15] In 1826, Ataíde sued the confraternity because he had not received full payment for the completed work. They countered that he had not fulfilled the commission as agreed upon, having used silver instead of gold, among other complaints. The contract, copied in the trial records, not only describes the iconography and style of the ceiling painting, but also details the steps involved in gilding, such as the number of layers of gesso and glue required for a suitably shiny finish (Menezes 2007b, pp. 201–2). Third parties were consulted to inspect the work and compile a detailed list of their observations, including any deviations from the contract and parts of the commission that were poorly executed (Menezes 2007b, p. 205). With such layers of supervision, unusual iconography, such as a Black Saint Gregory prominently displayed above the high altar, is likely to have either been requested by the patrons or, if proposed by artists, approved by the patrons.

This lawsuit also provides insight into the division of labor and role of assistants within a prominent white master's workshop. The confraternity members accused Ataíde of having left much of the work to his assistants while he worked on the Church of Nossa Senhora do Carmo in Ouro Preto. The testimonies clarify that the artists who carried out the commission were all or almost all Afro-Brazilian. A pardo painter, Jose Joaquim do Couto, testified that he worked on the high altar alongside Ataíde's own fourteen-year-old pardo son Francisco, two unnamed enslaved minors, and an apprentice named Raimundo whose race is not included in the testimony (Santiago 2008, p. 84). This account supports what scholars have long argued: although white master artists' names appear on contracts, much of Brazil's artistic heritage is the product of Afro-Brazilian labor and skill. In addition, the trial describes a division of labor in which the master artist not only delegated subsidiary tasks to Afro-Brazilian journeymen, assistants, apprentices, and enslaved artists, but also left them to work on their own with little supervision.

In addition to such trial records, master artists' testaments also often provide the names of enslaved artists. Judith Martins, Jose Luiz Mota Menezes, and Jaelson Bitran Trindade list dozens of enslaved artists documented in various regions of Brazil (Martins 1974; Menezes 2002; Trindade 2002). Trindade and Camila Santiago examined the working conditions of

enslaved artists and the legal and social limitations that they faced (Trindade 2002; Santiago 2008). Because of their legal status, archival records include relatively little information about enslaved artists. Often, their first name is all that is known. Documents may also include who enslaved them, their age, whether they were pardo, and born in Africa or criollo (in Brazil, the term describes people born locally to Black parents). The artists about whom we know more are the extraordinary exceptions who gained freedom and sometimes even master status. Joaquim Pinto de Oliveira Thebas was an enslaved pardo stonemason working in São Paulo who gained freedom, became a master, and even became a juror for the stonemasons guild (Lemos 2002). Unfortunately, much of his work has not survived to the present. Manuel da Cunha was an enslaved pardo artist who traveled to Lisbon, either while enslaved or after having been freed, and painted religious paintings and portraits in Rio de Janeiro. He has been sorely understudied, despite the survival of numerous paintings and at least some documentation. As Hannah Levy suggested in 1942, much further archival research is needed (Levy 1997, pp. 178–79). Well into the nineteenth century, when much had changed in the realm of artistic instruction, patronage, and exhibition, and the legal abolition of slavery was nearing, an enslaved pardo artist who took the name Antonio Benvenuto Celini was able to gain freedom, mastership, and employment at an art academy. He was born in 1847 and enslaved in the household of Jeronima Maria do Patrocinio Ramos in the state of Pernambuco (Cord 2014). At the age of nineteen, he won the third-place medal at the 1866 Exposicão Artistica e Industrial for a wooden sculpture and was presented with a letter of manumission at the award ceremony (Cord 2014, pp. 402–3). One detail to note about these three artists is that they were all pardo, demonstrating the role of colorism in artistic success, as Miguel Valerio discusses in this special issue.

The motivations and biases inherent in the written records and in scholarship create challenges for interpreting the dynamics of interpersonal relationships that occurred when free white or pardo artists, and enslaved artists were working together. In keeping with the fantasy of racial democracy, the systemic violence of such working relationships is often overlooked in favor of attributing behaviors to individual personality. For instance, Ataíde enslaved at least nine people throughout his career: Ana, Pedro, Manuel, Ambrosio, another Pedro, Maria, Victorino, Lucas, and Matheus (Campos 2002, pp. 77–79; Menezes 2007b, p. 213).[16] When he wrote his will, he was enslaving four people. Of those four, he stipulated that only Pedro and Ana would be freed after his death, but declined to free the two minors, Lucas and Matheus, the artists who worked on his commission for Nossa Senhora do Rosario dos Pretos in Mariana (Menezes 2007b, p. 213). Nevertheless, Campos argues that Ataíde was kind, emphasizing that he brought the enslaved members of his household with him to take communion (Campos 2002, pp. 77–79). Significantly, this idea of kind enslavers that is common among slavery apologists contrasts starkly with the descriptions of the pardo artist Aleijadinho's actions. Despite his political motivations and romantic dramatization, Rodrigo José Ferreira Bretas' account of Aleijadinho's life is largely accepted in scholarship and popular culture. Bretas describes the enslaved Mauricio as remarkably loyal, despite Aleijadinho beating him "rigorously" with a mallet (Bretas 1858, p. 4). In the case of Aleijadinho, this violence is attributed to his tortured state and erratic personality. The inherent violence of enslaving artists and forcing them to work is rarely mentioned to avoid tarnishing the reputations of the master artists and the cultural heritage of Brazil.

## 5. Conclusions

A comprehensive overview of the roles of Afro-Brazilians in Brazil's artistic patrimony includes a vast number of artists and artisans, as well as patrons. The artists are heterogeneous in terms of training, type of work, and social, legal, and professional status. There is a significant disparity in our knowledge of artists, with little known about most and much known about a select group. Given these challenges and limitations about what is known about artists, research on Afro-Brazilian patronage offers important contribu-

tions for understanding Black and pardo artistic agency. Black and pardo confraternities were major patrons of the arts, with impressive confraternity altars, chapels, and churches erected throughout Brazil, often employing the best artists available. As Miguel Valerio writes, "in a world that did not value black voices, Afro-Brazilians spoke instead through their confraternal monuments" (Valerio 2021, p. 241). Valerio's recent and forthcoming publications on Afro-Brazilian confraternities depart from attempts to see racial democracy visualized and instead examine how the iconographic program of a church as a whole served the purposes of the confraternity, allowing them to exert agency and express their corporate culture. For instance, in his interpretation of the church of Nossa Senhora do Rosário dos Pretos in Ouro Preto, Valerio examines not only the significance of the many Black saints, but also the white Saint Helena, concluding that the church's interior echoes the confraternity's mission of conversion (Valerio 2021).

Similarly, Jaelson Bitran Trindade contextualizes the depiction of a Black Saint Gregory within the church of Santa Efigênia to dispel some of the narratives that are not supported by evidence and to consider how the confraternity may have understood the painting (Trindade 2019). He examines print sources for the figure, considers Saint Gregory within the context of the four doctors and other aspects of the church's iconography, proposes that Saint Gregory was particularly important for the confraternity's religious message and, finally, that these Catholic messages did not prevent members of the confraternity from also associating saints with African religious figures. Careful consideration of the context in which artists worked and the heterogeneous communities that viewed artworks enables evidence-based interpretations.

Many of the artists that worked on major commissions were themselves members of Afro-Brazilian confraternities. Investigating the confraternities further enables an understanding of the culture and community within which artists were situated. Correlating supposedly African or Black styles, techniques, and iconography with Black and pardo people because of their race is misleading. Considering the context in which they lived and worked, their artistic training, their communities, and their relationships to religion, ritual, and visual culture enables plausible interpretations about what art meant to Afro-Brazilian artists and patrons, without biological essentialism. Such careful consideration allowed scholars in Fromont's edited volume on Afro-Catholic festivals to offer in-depth and nuanced interpretations of Catholic Central Africans' experiences on both sides of the Atlantic.

Fortunately, many regional archives are accessible to scholars in Brazil, and local history departments are teaching students and emerging scholars the necessary skills to conduct archival research. Archival research is an area in which scholars have made immense progress and continue to make important findings. For instance, Santiago's consultation of artists' testaments and the Ataíde trial allowed her to examine the role of enslaved artists (Santiago 2008).

Combining such archival research with a close examination of artworks may enable more artists' names to be included in art historical scholarship. For instance, scholars have attributed the ceiling painting of the chancel of Nossa Senhora do Rosario dos Pretos in Mariana to Ataíde's workshop, an attribution that the trial record confirms. Might the examination of further records and comparison with other works attributed to his workshop enable scholars to plausibly identify which artists were responsible for other ceiling paintings? For instance, the nave ceiling of the church of Santo Antonio in Santa Barbara has been attributed to Ataíde's workshop and was completed by 1822, the year before work began on the chancel ceiling in Mariana. Were Jose Joaquim do Couto, Ataíde's son Francisco de Assis Pacifico da Conceição, Lucas, Matheus, and the apprentice Raimundo also involved in that project? Furthermore, by the time of Ataíde's death in 1830, Lucas was 22 years old and Matheus 30, and the two were probably highly skilled artists (Menezes 2007b, p. 215). Further archival research may identify what happened to them after Ataíde's death.

A former president of Portugal, Mario Soares, wrote in his introduction to the exhibition *A Mão Afro-Brasileira*, "miscegenation was always an essential feature of Portuguese colonization, that transcended the mere extraction of riches from continents in which they settled, unlike other colonialisms."[17] Such narratives of Brazilian exceptionalism not only champion racial democracy, but are also prevalent in twentieth-century art historical publications. In part, the focus on mixed-race, and Aleijadinho in particular, was fueled by modernist desires to distinguish Brazil's colonial art from the art of Portugal and Europe as a whole (Bald 2001; Grammont 2008). Departing from the modernist ideal of originality, newer scholarship has investigated connections between Brazil's artistic heritage and Europe, especially in regard to the importation of prints, enabling more nuanced interpretations.

Acknowledging the white supremacist foundations of traditional art historical approaches enables new avenues of exploration. Recognizing that Black and pardo artists could choose to create artwork that was visually indistinguishable from the work of white artists encourages the examination of previously neglected work. Similarly, aspects of Afro-Brazilian patronage that scholars have not perceived as related to Africa are in need of further attention, such as the blue-on-white paintings on the chancel walls of the church of Santa Efigênia in Ouro Preto that I plan to address in a future publication on *chinoiserie* in Brazilian and Portuguese religious spaces. Lists of visible traces of Africanness or visible signs of the racial mixing of Brazil do not inform about an artwork's meaning, purpose, creation, or reception. For instance, in the sacristy of the convent of Santo Antonio in Paraiba, the painted representation of architecture is held up by two Atlantes painted in grisaille that are probably intended to be Black, with features that European artists associated with Africans—tightly curled hair, full lips, and small noses (Oliveira 2011, p. 111). These figures are in a servile position, nearly nude, and exerting effort to support the architecture above. Their meaning and function are very different than those of a painting of Saint Gregory the Great as a Black man. Departing from the myth of racial democracy enables scholars to examine the processes that brought about such representations without masking the violent circumstances in which Brazil's celebrated cultural heritage was produced.

**Funding:** This research received no external funding.

**Data Availability Statement:** Not applicable.

**Acknowledgments:** I would like to thank Mónica Dominguez Torres, Sabena Kull, and Miguel Valerio for engaging me in discussions on race in colonial Brazil over the years, and the anonymous reviewers for their insightful and helpful feedback.

**Conflicts of Interest:** The author declares no conflict of interest.

## Notes

1  "Toda desgraça que isso causou ao país, causou uma coisa boa, que foi a mistura, a miscigenação", diz Lula sobre vinda de negros escravizados para o Brasil," (Metrópoles 2023).

2  I will be referring to all Brazilians who were African or of African descent as "Afro-Brazilian." In keeping with Brazilian practice, "pardo" will refer to people with both Black and white parents or grandparents, while "Black" refers to those without known white ancestry.

3  For context on Freyre's writing, see (Alberto 2011, pp. 22–23).

4  Tupinamba featherwork created earlier in the colonial period has received significant attention. See, for instance, Amy Buono's numerous publications on the subject.

5  For historiographical discussions of the study of colonial, modern, and contemporary Afro-Brazilian art, see (Araujo 2002; Menezes 2018).

6  On nineteenth-century efforts to whiten Brazil's population, see the work of Patrícia Martins Marcos.

7  On the white supremacist violence of Bolsonaro's presidency, see (Perry 2020).

8  Within the baptismal records that I recorded during field research, the only mention of whiteness was for a "pai branco incognito" (unknown white father) who had impregnated a parda woman (*Baptisms* 1719–1736, folio 69r).

9    "Uma fina camada de pó marrom escuro recobria todo o medalhão, incluindo as figuras de Nossa Senhora da Porciúncula, do Rei David e dos anjos." (Coelho 2007, p. 88).

10   "Os traços faciais de Maria aproximam-se do tipo físico mestiço, com lábios carnudos e cabelos ondulados", (Oliveira 2011, p. 106).

11   The second child's baptismal certificate has not been identified. She was listed as two years old in 1814. (Campos 2007, p. 77).

12   The baptismal certificate for their first child, Francisco, was written on 23 October 1809. (Campos 2007, p. 74, note 21).

13   Lisa Voigt, Cécile Fromont, and Junia Ferreira Furtado wrote about Brazilian festivals.

14   As Hélio Menezes discusses, correlating visual characteristics of an artwork to the race of the artist persists in scholarship on contemporary Brazilian art (Menezes 2018, p. 577).

15   Some of the records from the lawsuit are transcribed in (Menezes 2007b, pp. 199–210). The original manuscript is *Manoel da Costa Ataíde*, 1826, codex 239, auto 5972, 2nd ofício, Arquivo da Casa Setecentista de Mariana which is referenced and discussed in (Santiago 2008, p. 84).

16   As Camila Santiago suggests, the second record of a Pedro from Angola may have been the same person if the scribe accidentally transposed the age. (Santiago 2008, p. 81, n. 12).

17   "A miscigenação foi sempre um traço essencial do colonialismo português, que extrapolou a mera exploração das riquezas dos continentes em que se implantou, diferentemente dos demais colonialismos", (Soares 2002).

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
