# Peer review of "Racial Democracy, Visibility, and the History of Colonial Brazilian Art"

_arts, 2023_

Round 1

Reviewer 1 Report

Great article. Thank you for this fantastic contribution to this debate. Just correct year 1821 (1823 - 1809 = 1814).

Author Response

Thank you!

Reviewer 2 Report

The article makes important historiographic and critical interventions into colonial Brazilian art history. However, it is missing important methodological framing and lacks sufficient visual analysis of several of the art works discussed.

Some specific notes below:

-The conceptualizing of race as biological needs re-framing. This issue of if/whether “race” was considered biological or social (or both) needs to be more fully explored. “Biology” is the very beginning of the title yet not mentioned in the introductory paragraph. Does this relate to Linnean taxonomy? 

-Conversely, because sexual violence is foregrounded in the essay, it probably should be mentioned in the title. 

-Much of the theory used is decolonial theory, and needs to be explained and cited as such. Particularly for the section about the lawsuit, look to the work of Ananda Cohen-Aponte for a model on this. 

-General terms like “problematic” need to be more specifically defined and analyzed.

-In the section on Ataide’s Virgin, skin color needs to be specifically described. If skin color cannot be accurately used as a marker of race in the interpretation of the image (due to degradation of materials, damage, deliberate repaintings etc), this needs to be expressly discussed.

The article needs rigorous editing to pick up typos, syntax errors, and unclear language.

Author Response

--The conceptualizing of race as biological needs re-framing. --Thank you for this comment. Titles are not my strong suit and I agree that the mention of "biology" is not appropriate. I have removed that part of the title and edited language for clarity throughout. 

-Conversely, because sexual violence is foregrounded in the essay, it probably should be mentioned in the title. --I will keep this in mind and discuss it with the editors. Readers familiar with criticisms of racial democracy will be aware of its foundations on sexual violence. My hope is that readers will come to recognize the inherent violence of racial democracy rather than to see racial democracy and sexual violence as separate.  

-Much of the theory used is decolonial theory, and needs to be explained and cited as such. Particularly for the section about the lawsuit, look to the work of Ananda Cohen-Aponte for a model on this. --I looked at several of Cohen-Aponte's publications and am not certain how to incorporate her work here. Unlike the politically and racially charged context of testimonies regarding an Indigenous rebellion, the testimonies gathered for the Ataide case would presumably have been less fraught with coercion and leading questions. 

 -General terms like “problematic” need to be more specifically defined and analyzed.--Yes, thank you. I altered the wording. 

-In the section on Ataide’s Virgin, skin color needs to be specifically described. If skin color cannot be accurately used as a marker of race in the interpretation of the image (due to degradation of materials, damage, deliberate repaintings etc), this needs to be expressly discussed. -- I have added an in-depth discussion of this topic, of normative whiteness, conservation, and such. 

Thank you very much for your help!

Reviewer 3 Report

This is an interesting article that raises important questions related to the interpretation of colonial religious art in Brazil, principally by investigating the apparent biases of art historians. According to the author, these scholars tend to highlight the visible signs of Afro-Brazilian cultures in colonial religious art as evidence of harmonious racial mixing, an ideology prevalent in Brazil since the 1930s. Moreover, following Dean and Leibsohn, the author argues that they usually associate the presence of visible African traces to the hands of Afro-Brazilian artists, which typecast the latter, overlooking issues such as artistic agency, artistic training and patronage. This oversight, contends the author, is based on long-standing white supremacist tropes that interpret artworks according to the race of the artist.

While this article would make a relevant contribution to the special issue ‘Black Artists in the Atlantic World’, especially because of its focus on the Luso-Brazilian context, its argument needs to be developed and nuanced according to the suggestions/queries below.

1.     The use of the expression ‘white supremacist’ to refer to white power structures that perpetuate racialised violence needs to be explained. It would be helpful to spell out the difference between the way the expression is being used in the article, as a global phenomenon, and the localised identity-based particularity of white supremacy. Aisha M. Beliso-De Jesús and Jemima Pierre’s ‘Introduction’ in the Special Section: Anthropology of White Supremacy, American Anthropologist 122: 1 (2020) 65-75 might be helpful.

2.     In order to nuance the generalisation about Brazilian art historians perpetuating the myth of racial democracy, it would be useful to consider Mário de Andrade’s appraisal of Padre Jesuíno de Monte Carmelo’s colonial religious art in the state of São Paulo. A mulato intellectual himself, Mário de Andrade was particularly attuned to Padre Jesuíno’s agency and challenges he had to overcome as a mulato artist to achieve recognition by his patrons. See Mário de Andrade, Padre Jesuíno de Monte Carmelo (Rio de Janeiro: Nova Fronteira, 2012).

3.     The title of the article ‘The Limits of Biology’ is misleading, because processes of racialisation are not only related to biology, and this is not discussed by the author. I would suggest keeping the title as ‘Racial Democracy, Visibility, and the History of Colonial Brazilian Art’.

4.     The discussion of Ataíde’s whiteness as a justification for his Virgin being white despite her facial features not matching canonical representations of a white face (lines 125-127) is more complex than the author expresses it. It would be relevant to consider that what is understood as ‘white’ in Brazil – a very mixed society – is already different from the Eurocentric pictorial conventions of whiteness, and therefore while Ataíde’s intention might have been to represent the Virgin as white, using his ‘standard method of painting faces’, the Virgin’s face would be seen by art historians as a mixed-race face. The point is that to be considered ‘white’ in Brazil is not just a question of racial features, but of social position as well. This is not to contradict the valid argument that the author makes about the misreading by art historians of the representation of a mixed-race Virgin as a celebration of racial democracy, but to add some complexity to this interpretation (the same should be considered for the argument in lines 139-142 and 160-165) .

5.     Reference for information in lines 406-417 should be added.

6.     Is there any document evidencing that the representation of a Black Saint Gregory has been requested by the patrons (lines 430-432)?

Minor corrections:

a.     Line 37: complexities of Brazil’s

b.     Line 62: more inclusive interpretative modules

c.     Line 68: Afro-Brazilian people

d.     Line 81: studied for a long time [instead of ‘for much time’]

e.     Line 82: the 1930s, the sociologist Gilberto Freyre

f.      Throughout the article, but in p. 3 in particular, there is a lot of repetition of sentences such as ‘In addition to romanticizing sexual violence, art history emphasizes that harmonious racial mixing was made visible in art of the colonial period with the near ubiquitous depiction of mixed-race individuals in prestigious religious artworks.’ This should be avoided.

g.     Line 111: both Black and white parents

h.    Line 230: Conduru

i.      Line 241: Efigenia [to be consistent]

j.      Line 336: Conceição

k.     Line 361: St Augustine

l.      Line 384: this section

m.   Line 421: azulejos. In 1826 Ataíde

n.    Line 470: Exposição

Author Response

The use of the expression ‘white supremacist’ to refer to white power structures that perpetuate racialised violence needs to be explained. It would be helpful to spell out the difference between the way the expression is being used in the article, as a global phenomenon, and the localised identity-based particularity of white supremacy. Aisha M. Beliso-De Jesús and Jemima Pierre’s ‘Introduction’ in the Special Section: Anthropology of White Supremacy, American Anthropologist 122: 1 (2020) 65-75 might be helpful. -- Thank you for this recommendation. It was very helpful both for this article and another project. 

In order to nuance the generalisation about Brazilian art historians perpetuating the myth of racial democracy, it would be useful to consider Mário de Andrade’s appraisal of Padre Jesuíno de Monte Carmelo’s colonial religious art in the state of São Paulo. A mulato intellectual himself, Mário de Andrade was particularly attuned to Padre Jesuíno’s agency and challenges he had to overcome as a mulato artist to achieve recognition by his patrons. See Mário de Andrade, Padre Jesuíno de Monte Carmelo (Rio de Janeiro: Nova Fronteira, 2012).---Thank you for this comment. I agree that my hurried writing did not appropriately represent the field of research on the Brazilian Baroque. I have added several paragraphs that provide a more nuanced overview of the state of the field. 

The title of the article ‘The Limits of Biology’ is misleading, because processes of racialisation are not only related to biology, and this is not discussed by the author. I would suggest keeping the title as ‘Racial Democracy, Visibility, and the History of Colonial Brazilian Art’. ---Yes,  thank you. Titles are always a challenge for me. This suggestion is very helpful. 

The discussion of Ataíde’s whiteness as a justification for his Virgin being white despite her facial features not matching canonical representations of a white face (lines 125-127) is more complex than the author expresses it. It would be relevant to consider that what is understood as ‘white’ in Brazil – a very mixed society – is already different from the Eurocentric pictorial conventions of whiteness, and therefore while Ataíde’s intention might have been to represent the Virgin as white, using his ‘standard method of painting faces’, the Virgin’s face would be seen by art historians as a mixed-race face. The point is that to be considered ‘white’ in Brazil is not just a question of racial features, but of social position as well. This is not to contradict the valid argument that the author makes about the misreading by art historians of the representation of a mixed-race Virgin as a celebration of racial democracy, but to add some complexity to this interpretation (the same should be considered for the argument in lines 139-142 and 160-165) .---Thank you. I have added some paragraphs that clarify the definitions of whiteness at the time and have revised my discussion of this painting. 

Reference for information in lines 406-417 should be added. --Done! 

Is there any document evidencing that the representation of a Black Saint Gregory has been requested by the patrons (lines 430-432)? Unfortunately, not that I know of. I have reworded that sentence to hopefully make my intentions clearer. 

Thank you very much for your help! Your feedback is greatly appreciated. I also fixed the typos that you found.